Carbon content and other soil properties of near-surface peats before and after peatland restoration

Hammerich Jenny 1 2 jenny.hammerich@hnee.de
Schulz Corinna 1 3
Probst Robert 1
Lüdicke Thomas 4
Luthardt Vera 1
1 Faculty of Landscape Management & Nature Conservation, Eberswalde University for Sustainable Development , Eberswalde, Brandenburg , Germany
2 Center of Methods & Faculty of Sustainability, Leuphana University , Lüneburg, Lower Saxony , Germany
3 Institute of Botany and Landscape Ecology, Greifswald University , Greifswald, Mecklenburg-Western Pomerania , Germany
4 UWEG Engineers & Analytics GmbH , Eberswalde, Brandenburg , Germany
Karabiniuk Mykola
Electronic publication date: 2024 Apr 18
Publication date: 2024
Volume: 12
Electronic Location ID: e17113
Received 2023 May 22; Accepted 2024 Feb 26
Copyright: © 2024 Hammerich et al.
Copyright year: 2024
Copyright holder: Hammerich et al.
License: This is an open access article distributed under the terms of the Creative Commons Attribution License, which permits unrestricted use, distribution, reproduction and adaptation in any medium and for any purpose provided that it is properly attributed. For attribution, the original author(s), title, publication source (PeerJ) and either DOI or URL of the article must be cited.
License URL: https://creativecommons.org/licenses/by/4.0/

Keywords: Mire, Rewetting, pH value, Nitrogen content, C/N ratio, Dry bulk density

Funding: State Environmental Agency of Brandenburg This work was supported by the State Environmental Agency of Brandenburg. The funders had no role in study design, data collection and analysis, decision to publish, or preparation of the manuscript.

==============================
Peatland restoration usually aims at restarting the peatlands’ function to store carbon within peat. The soil properties of the near-surface peat can give a first understanding of this process. Therefore, we sampled pH value, total organic carbon content (TOC), total nitrogen content (TN), C/N ratio as well as dry bulk density (BD), and describe the structure of near-surface peats in six restored fens in North-East Germany before (2002–2004) and after (2019–2021) restoration. Before restoration, the study sites showed peat degradation to various extents in their near-surface peats. pH values remained relatively stable over time. Comparing the degraded peat horizons, TOC increased significantly in four study sites, ranging from 35.7% to 47.8% in 2002–2004 and from 42.5% to 54.0% in 2019–2021. TN varied from 1.5% to 3.5% in 2002–2004 and from 1.8% to 3.2% in 2019–2021, but changes were only significant in one site, showing a slight decrease. In three sites, the increase in C/N ratio was significant, indicating lower nutrient availability. BD ranged from 0.08 to 0.48 g/cm3 in 2002–2004 and from 0.10 to 0.16 g/cm3 in 2019–2021, decreasing significantly in four sites. The structure of the degraded peat horizons changed after restoration to a more homogenous, sludge mass with larger re-aggregates. In three sites, new peat moss peat layers above the degraded soil horizon were present in 2019–2021, with a mean thickness of 6.8 to 36.1 cm. The structure was comparable to typical, slightly decomposed peat moss peat. Our findings suggest that within about 17 years after fen restoration, and thereby a water table rise close to surface, TOC of the near-surface peats increased to values that are typical for undisturbed peatlands. This indicates that restoration can lead to the re-establishment of peatlands as potential carbon sinks, with TOC within the near-surface peat as one key factor in this process. Further, we assume that the decrease in nutrient availability, decrease of BD, and new, undisturbed peat layers can favor the establishment of mire-specific biodiversity and support ecosystem services similar to near-natural mires.

Introduction

Peatlands are a major global carbon store, containing more carbon than any other terrestrial ecosystem in vegetation and peat (Joosten et al., 2016). They are characterised by the presence of a naturally accumulated layer of peat, which is sedentarily accumulated material whose dry mass is at least 30% dead organic material, meaning 15% of total organic carbon, according to Joosten & Clarke (2002) and the German definition of mires (Ad-hoc-AG Boden, 2005). Due to the water saturated and therefore anaerobic milieu in the soil, the decomposition of the on-site dead biomass is very slow and organic matter in the form of plant remains is stored as peat (Zeitz, 2014). The anaerobic conditions usually lead to the emission of methane (Lai, 2009). If peatlands are drained, peat formation stops. Oxygen enters the near-surface peat and secondary pedogenetic processes, such as humification and mineralisation are initiated, resulting in the release of the previously stored carbon and nitrogen as greenhouse gases carbon dioxide (CO2) and nitrous oxide (N2O) (Zeitz, 2016).

Nearly 50% of the remaining European peatland area is degraded, in particular due to drainage for agriculture and forestry (Joosten & Tanneberger, 2017). In the federal state of Brandenburg in North-East Germany, where the study area is located, only about 2% of the remaining 163.000 ha peatland area is in a natural state (LfU (Landesamt für Umwelt Brandenburg), 2016; Luthardt & Zeitz, 2014). This results in estimated total emissions of 6.3 million t CO2-Eq. per year, accounting for 11% of the state’s annual greenhouse gas emissions (Reichelt, 2021). The once existing specific biodiversity of mires (peatlands with a vegetation that actively forms peat (Joosten et al., 2017)) is lost, resulting for example in the decline of species, as 62% of mire-specific vascular plants in Brandenburg are listed as highly endangered, at risk of extinction or extinct (cf. LUA (Landesumweltamt Brandenburg), 2006; Luthardt, 2014; Hammerich et al., 2022). Ecosystem functions, such as stabilization of the landscape water balance or functioning as flood retention areas, water storage basins, groundwater nourishment areas and regulators of microclimate, are changed completely for the benefit of provisioning services (Luthardt & Wichmann, 2016). In order to reduce CO2 emissions and restore the function as carbon sinks, as well as biodiversity and other ecosystem functions of mires, peatland restoration efforts have risen (Bonn et al., 2016). Peatland restoration activities in Europe started in the late 20th century, but are estimated to be insufficient and slowly progressing (Roe et al., 2019; Tanneberger et al., 2021; Greifswald Mire Centre and Wetlands International, 2022).

The near-surface peat is the location where carbon is accumulated for transfer into the long-term carbon store. Researching changes in carbon content in near-surface peat is therefore one key factor within this process (cf. Tolonen & Turunen, 1996; Price et al., 2016).

With regard to peatlands as global carbon store, total organic carbon content (TOC) appears to be the most relevant soil parameter. TOC accounts for about 58% of the total organic matter formed by plant remains (Montanarella, Jones & Hiederer, 2006). Klingenfuß et al. (2014) specify this relation for peat soils in North-East Germany, giving values from 49% to 58%, depending on peat type.

The carbon to nitrogen (C/N) ratio is a key parameter regarding the trophic conditions in peatlands. The smaller the ratio, the more nutrients are plant-available (Succow & Stegmann, 2001).

In addition to organic matter, peat soils consist of varying proportions of mineral matter, also referred to as ‘ash’ (Succow, 1988). This can involve allochthonous mineral matter that is carried in by surface waters, contributed by some form of soil amendment associated with agriculture or is attributable to the loss of organic matter during drainage-based peat mineralisation (Klingenfuß et al., 2014).

Dry bulk density (BD) is an essential physical parameter and necessary to estimate peatland carbon stocks. Higher BD values are associated with a lower carbon content and increase from near-natural sites to sites impacted by drainage (Chapman et al., 2017; Wittnebel, Tiemeyer & Dettmann, 2021).

Due to the relatively short time period of peatland restoration activities and the restricted availability of long-term monitoring data (cf. Andersen et al., 2016), there is only little research focusing on carbon gained over time in near-surface peats, especially regarding fens as geogenous peatlands in contrast to rainwater-fed bogs (Kotowski et al., 2016; Mrotzek et al., 2020).

In order to fill the knowledge gap concerning the re-establishment of restored peatlands, especially fens, as potential carbon sinks, we report changes in the soil parameters pH value, TOC, total nitrogen content (TN), C/N ratio, BD and structure of near-surface peats of six restored fens in North-East Germany before (2002–2004) and after (2019–2021) restoration measures were initiated. We refer to different hydrological and ecological site conditions as well as different soil degradation stages.

Materials and Methods

Study sites

The study sites, six fens in the nature park ‘Stechlin-Ruppiner Land’ (Table 1), are located in the federal state of Brandenburg, North-East Germany in the nemoral zone. The landscape formed in the late Pleistocene and is therefore characterised by large differences in height of the various landscape elements (high relief energy) and diversity of peatland types (Luthardt et al., 2002). The ‘hydrogenetic peatland type’ provides information on the hydrological conditions of formation and the resulting composition of the peatland. Thereby, Succow (1988) distinguishes between ombrogenous bogs and seven different geogenous fen types, a classification largely congruent to the suggestions given by Joosten & Clarke (2002) for a global hydrogenetic classification of peatlands. The study sites comprise three terrestrialisation peatlands (Beerenwiese, Jägerwiese, Müllerwiese), one percolation peatland (Boberowseewiese), one water rise peatland (Seggenkuten) and one kettle hole peatland (Teufelsseemoor) (Table 1). The specifications regarding the ‘ecological peatland type’ given in Table 1 also follow Succow (1988). These types provide information on the chemical quality of the feeding water in terms of trophic conditions (nutrient availability) and base saturation (acidity), which lead to the establishment of different characteristic plant communities under undisturbed conditions. Regarding the trophic conditions, which are quantified by the C/N ratio of the peat, Succow (1988) distinguishes oligotrophic (>33), mesotrophic (33–20) and eutrophic (<20–10) peatlands. The base saturation is deviated from the pH value of the peat: acidic (<4.8), subneutral (4.8–6.4) and calcareous (>6.4). The study sites comprise a variety of ecological types: four of them are meso- to eutrophic acidic (Jägerwiese, Müllerwiese, Seggenkuten, Teufelsseemoor), one is meso- to eutrophic subneutral (Beerenwiese), and one is eutrophic subneutral (Boberowseewiese).

Table 1 Study sites, characteristics and restoration measures implemented in the EU-Life-Project ‘Restoration of clear water lakes, mires and swamp forests of Lake Stechlin’.

Site and size (ha)	Ecological and hydrogenetic peatland type	Degradation of near-surface peat	Main restoration measures	Area share (%) of water table typical for near-natural mires	Trend of water table after restoration	Dominant vegetation after restoration on soil plots	
Before restoration (2002–2004)	After restoration (2019–2021)	
Beeren-wiese
(10.5)	Mesotrophic to eutrophic subneutral terrestrialisation peatland	Earthification	Water table rise in adjacent lake (+30 cm) (achieved in 2007)	49	51	Rising	Sedges (Carex spec.)	
Boberow-seewiese (6.3)	Eutrophic subneutral percolation peatland	Murshi-fication	Partial closure of drainage ditch, water table rise in adjacent lake (+20 cm) (2003)	5	68	Rising	Sedges (Carex spec.)	
Jägerwiese
(1.4)	Mesotrophic to eutrophic acidic terrestrialisation peatland	Beginning earthi-fication	Closure of drainage ditch (2003)	3	51	Rising	Sedges (Carex spec.) with few peat mosses (Sphagnum spec.)	
Müller-wiese
(2.0)	Mesotrophic to eutrophic acidic terrestrialisation peatland	Earthification	Closure of drainage ditch (2003)	15	15	Rising	Peat mosses (Sphagnum spec.)	
Seggen-kuten
(0.17)	Mesotrophic to eutrophic acidic water rise peatland	Beginning earthi-fication	Removal of marginal spruces, stocking reduction in the above ground catchment area (2003)	45	100	Rising	Peat mosses (Sphagnum spec.)	
Teufels-seemoor
(0.95)	Mesotrophic to eutrophic acidic kettle hole peatland	Beginning earthi-fication	Closure of drainage ditch, pine curling, removal of marginal spruces (2003)	24	100	Rising	Peat mosses (Sphagnum spec.)	
Note:

The data is modified from Luthardt et al. (2021). The classification of ecological and hydrogenetic peatland type follows Succow (1988), classification of peat degradation according to Schulz et al. (2019).

The long-term annual rainfall in the study area is 528 mm, while the long-term annual mean temperature is 9.3 °C (period 2002–2021). In comparison to the 30-year average from 1961–1990, the temperature increased by 1.2 K and the precipitation decreased by 59 mm (DWD (Deutscher Wetterdienst), 2022). The climatic water balance is negative, so that the water content of the peatlands is always dependent on inflowing water from the surrounding environment (Luthardt & Zeitz, 2014).

Until 2002, all six fens were drained, most of them directly by drainage ditches, some also by indirect drainage due to evaporation-intensive stocking of conifers in the aboveground catchment area or lowering of the water table of adjacent lakes. Water tables in all fens were several tens of centimeters below peatland surface. Consequently, near-surface peats in all sites exhibited different stages of oxygen-induced soil degradation. The natural peat structure changes gradually to a crumb or even fine granular structure with substantially changed soil properties (Ilnicki & Zeitz, 2003). In German soil classification, moderate drainage leads to the development of ‘earthified’ peat, characterized by a crumb grain structure resembling garden mold. Under intensive drainage, aeration and ongoing degradation, the crumb structure subsequently changes into a structure of fine granular soil particles, referred to as ‘murshified’ peat (Ad-hoc-AG Boden, 2005; Schulz et al., 2019).

Between 2002 and 2004, several restoration measures were initiated as part of the EU-Life-Project ‘Restoration of clear water lakes, mires and swamp forests of the Lake Stechlin’ (Project-Ident: LIFE00 NAT/D/007057) (Luthardt et al., 2002, 2021), for details of restoration measures see Table 1. The water table was continuously measured monthly since June 2003 in the center of each fen in a permanently installed water level tube with a water level meter. All sites exhibit a water table rise following restoration (Fig. 1), due to the increased water retention as a consequence of the different restoration measures. A maximum was reached between 2011 and 2012, after years with high precipitation (2007, 2010, 2011). The percentage of area with a water table typical for near-natural mires (annual mean at least at peatland surface or higher) increased distinctly in four of the six sites about 15 years after restoration (Table 1). The highest increase was recorded in Teufelsseemoor with a peak water table up to 75 cm above surface, leading to a temporary shallow water body. In contrast, Müllerwiese is the only site where the water table is mainly still below peatland surface and the closure of the drainage ditch could not yet compensate for decades of drainage. Further, potentially peat-forming plant species which are adapted to high water tables, mainly sedges (Carex spec.) in Beerenwiese, Boberowseewiese, Jägerwiese or mainly peat mosses (Sphagnum spec.) in Müllerwiese, Seggenkuten, Teufelseemoor, could re-establish or expand (Fig. 2) (Luthardt et al., 2021).

Figure 1 Water tables in the study sites from 2003 (before restoration) to 2019 (cm related to peatland surface = 0).

Figure 2 Photo comparison of study sites before and after restoration.

Soil sampling

The field research was approved by the administration of nature park ‘Stechlin-Ruppiner Land’ as representative of the State Environmental Agency of Brandenburg. Soil data collection before restoration was conducted in 2002 and 2004 (Beerenwiese, Boberowseewiese, southern part of Seggenkuten, Teufelsseemoor in 2002; Jägerwiese, Müllerwiese, northern part of Seggenkuten in 2004). On each site, we installed two soil measurement fields with five sampling plots, respectively, giving ten replicas per site. Sampling plot locations were recorded via precision-GPS (LEICA GS 50). At each plot, we collected soil samples in the uppermost homogenous soil horizon, the degraded near-surface peat in a depth of 0–20 cm. Samples were taken volumetrically with sample rings of 100 cm3 volume and additionally a non-volumetrical bag sample of about 500 g at the same depth for further analysis.

Repeated, analogous sampling was conducted in 2019 for Beerenwiese, Boberowseewiese, Jägerwiese, and in 2021 for Müllerwiese, Seggenkuten and Teufelsseemoor. In some of the peat moss-dominated plots, we found a newly grown peat moss peat layer above the degraded peat, which formed the original surface layer in 2002–2004. In all cases, the newly formed peat moss peat was clearly distinguishable from the degraded peat layer below by color and structure (compare Fig. 3), appearing as a sharp border. While sedge roots as peat forming plant parts grow vertically into existing peat and form ‘displacement peat’, peat mosses grow upwards and form a new layer of peat moss peat above the existing peat when dying (Michaelis, Mrotzek & Couwenberg, 2020). In plots with a newly grown peat moss peat layer, we additionally collected soil samples from this new layer. The comparative samples to 2002–2004 were collected in the layer below, which corresponds to the original surface layer in 2002–2004. For Teufelsseemoor, volumetrically sampling was not possible due to the distinct fluffiness of the newly formed peat, which did not allow cutting without distorting its shape.

Figure 3 Representative peat monoliths of displacement peat and newly formed peat moss peat.

(A) wet and re-aggregated displacement peat formed by ingrowing and dying sedge roots in the degraded near-surface peat after restoration. (B) wet and re-aggregated degraded near-surface peat after restoration with 10 cm of newly formed peat moss peat on top, distinguishable by colour and structure

Laboratory analysis of soil parameters

Analysis of soil parameters were conducted at the analytical laboratory of Eberswalde University for Sustainable Development, Germany.

The determination of pH value was conducted according to German Institute for Standardisation (2005): composition of air-dry samples (dried at 40 °C in a forced-air oven) with calcium chloride solution and measurement after 1 h with an electrode (WTWMulti1970i).

Determination of BD was carried out according to German Institute for Standardisation (2017): samples were dried at 105 °C in a forced-air oven to constant mass. BD (g/cm³) was calculated by dividing the dry mass (g) by the volume of the sample ring (100 cm³) (Chambers, Beilman & Yu, 2011).

Determination of total carbon (TC) content was conducted according to German Institute for Standardisation (2022) and TN according to German Institute for Standardisation (1995): samples were dried at 40 °C and 1 to 2 g substrate were grounded in a fine mill to powder and dried again at 105 °C as pretreatment for dry combustion (1,200 °C) with an elemental analyzer (LECO Trumac CNS). Due to low pH values and conduction of a carbonate pre-test with 10% hydrochloric acid, all samples were to be regarded as carbonate-free. Therefore, the measured TC corresponds to TOC.

Statistical analysis of soil parameters

For each site and each time, mean values and standard deviations were calculated for the soil parameters pH value, TOC, TN, C/N ratio and BD. The Wilcoxon-test was used to detect possible significant differences of soil properties regarding time span before and after restoration, since not all samples were normally distributed, indicated by the Shapiro-Wilk-test. All statistics were carried out with R (R Core Team, 2023), package ‘psych’ (Revelle, 2023) and ‘EnvStats’ (Millard, 2013). Figures were drawn with R-package ‘ggplot2’ (Wickham, 2016).

Results

All mean values of soil parameters described in the following refer to the comparison of the degraded peat layer before and after restoration about 17 years later. The parameters are given in detail in Table 2 and Figs. 4 to 8.

Table 2 pH value, TOC, TN, C/N ratio and BD of the degraded near-surface peats before (2002–2004) and after (2019–2021) restoration.

Site	Sampling year	Number of replicas	pH value	TOC-total organic carbon (%), mean (SD)	TN-total nitrogen (%), mean (SD)	C/N ratio, mean (SD)	BD-dry bulk density (g/cm3), mean (SD)	
Beeren-wiese	2002–2004	10	5.44 (0.11)	36.18 (1.79)	3.54 (0.53)	10.44 (1.69)	0.18 (0.03)	
2019–2021	10	5.22 (0.18)*	43.06 (2.26)*	3.23 (0.37)*	13.50 (1.93)*	0.13 (0.03)*	
Boberow-seewiese	2002–2004	10	5.01 (0.22)	35.67 (1.72)	3.13 (0.52)	11.69 (1.99)	0.48 (0.13)	
2019–2021	10	5.30 (0.26)	42.49 (4.40)*	3.05 (0.45)	14.36 (3.91)*	0.15 (0.05)*	
Jägerwiese	2002–2004	10	3.16 (0.22)	46.23 (1.76)	1.50 (0.43)	33.12 (9.31)	0.08 (0.01)	
2019–2021	10	3.26 (0.12)	50.36 (1.31)*	1.89 (0.29)	27.36 (5.07)	0.10 (0.02)	
Müller-wiese	2002–2004	10	3.53 (0.33)	47.79 (2.34)	2.02 (0.37)	24.59 (5.72)	0.12 (0.03)	
2019–2021	10	3.56 (0.22)	47.65 (6.61)	2.32 (0.27)	20.68 (3.12)	0.14 (0.07)	
Seggen-kuten	2002–2004	10	3.58 (0.25)	44.79 (5.49)	2.81 (1.05)	17.75 (5.51)	0.21 (0.07)	
2019–2021	10	3.58 (0.18)	47.61 (3.09)	2.36 (0.18)	20.28 (1.92)	0.16 (0.03)*	
Teufelssee-moor	2002–2004	10	3.26 (0.13)	38.88 (2.46)	1.88 (0.70)	22.70 (6.44)	0.28 (0.07)	
2019–2021	10	3.09 (0.13)	53.95 (1.23)*	1.79 (0.18)	30.48 (3.69)*	0.14 (0.02)*	
Note:

Mean values, standard deviations (SD, in brackets) and respective number of replicas for the soil parameters pH value, TOC, TN, C/N ratio and BD of the degraded near-surface peats of the study sites before (2002–2004) and after (2019–2021) restoration. Asterisks indicate significant changes (p < 0.05) between both time series.

Figure 4 pH value of the degraded near-surface peats before (2002–2004) and after (2019–2021) restoration.

Figure 5 TOC-total organic carbon (%) of the degraded near-surface peats before (2002–2004) and after (2019–2021) restoration.

Figure 6 TN-total nitrogen (%) of the degraded near-surface peats before (2002–2004) and after (2019–2021) restoration.

Figure 7 C/N ratio of the degraded near-surface peats before (2002–2004) and after (2019–2021) restoration.

Figure 8 BD-dry bulk density (g/cm3) of the degraded near-surface peats before (2002–2004) and after (2019–2021) restoration.

In five of the six sites, pH values remain relatively stable over time, ranging from 3.1 to 3.6 in the acidic fens and from 5.0 to 5.4 in the subneutral ones. Only in Beerenwiese, there is a slight but significant decrease from 5.4 to 5.2 after restoration (Fig. 4).

TOC between sites ranges from 35.7% to 47.8% in 2002–2004 before restoration and from 42.5% to 54.0% in 2019–2021 after restoration. TOC increases distinctly in five sites (four significantly, thereof three sedge-dominated, one peat moss-dominated) and remains stable in Müllerwiese. The highest gain (15.1%) is recorded in Teufelsseemoor (Fig. 5).

Differences in TN are less distinctive and only significant for one site (Beerenwiese: decrease from 3.5% to 3.2%). The TN decreases in four sites, increases in two sites and ranges from 1.5% to 3.5% in 2002–2004 and from 1.8% to 3.2% in 2019–2021 after restoration (Fig. 6).

The C/N ratio increases in four sites (three significantly), indicating a lowered nutrient availability. These four sites are the same with a recorded decrease in TN (Beerenwiese, Boberowseewiese, Seggenkuten, Teufelsseemoor) (Fig. 7).

BD between sites ranges from 0.08 to 0.48 g/cm3 in 2002–2004 and from 0.10 to 0.16 g/cm3 in 2019–2021. It significantly decreases in four sites (Beerenwiese, Boberowseewiese, Seggenkuten and Teufelsseemoor), most remarkably in Boberowseewiese with the formerly highest degree of peat degradation (murshification) from 0.48 g/cm3 before rewetting to 0.15 g/cm3 afterwards. There are slight and not significant increases in Jägerwiese (0.08 to 0.10 g/cm3) and in Müllerwiese (0.12 to 0.14 g/cm3) (Fig. 8).

The typical structure of the formerly degraded peat horizons (crumb grain structure for earthified peat and fine granular structure for murshified peat according to Schulz et al., 2019) changed to a more homogenous, sludged mass with larger re-aggregates in the rewetted peats.

In three sites, the formation of a new peat layer formed by peat mosses above the degraded soil horizon was recorded in 2019–2021 (nine soil plots in Müllerwiese, four in Seggenkuten and all ten plots in Teufelseemoor (Table 3). The mean thickness of this layer ranges from 6.8 cm in Seggenkuten to 36.1 cm in Teufelsseemoor. This is equivalent to mean accumulation rates of 0.38–2.01 cm/y considering the time since restoration measures in 2003. The structure of the newly formed peat moss peat is analogous to slightly decomposed peat moss peat, as described by Schulz et al. (2019): loosely bedded, felty and well preserved moss plants with a characteristic straw yellow to light brown color (Fig. 3). Mean TOC in new peat is comparable to TOC in older peat at two of the three sites after restoration, whereas in Teufelsseemoor it is significantly higher in the new peat (51.7%). TN is significantly lower in one site (Müllerwiese), significantly higher in one site (Teufelsseemoor) and remains stable in Seggenkuten. Values for C/N ration act reciprocal (Table 3). BD in this new peat is lower in all measured sites, but only significantly in Müllerwiese.

Table 3 Thickness, pH value, TOC, TN, C/N ratio and BD of the newly formed peat moss peat above the degraded peat of the study sites Müllerwiese, Seggenkuten and Teufelsseemoor in 2021.

Site	Number of replicas	Thickness of newly formed peat moss peat above degraded peat (cm), mean (SD)	pH value	TOC-total organic carbon (%), mean (SD)	TN-total nitrogen (%), mean (SD)	C/N ratio, mean (SD)	BD-dry bulk density (g/cm3), mean (SD)	
Müller-wiese	9	15.78 (5.89)	3.24 (0.10)*	48.83 (0.51)	1.33 (0.27)*	37.96 (7.21)*	0.06 (0.02)*	
Seggen-kuten	4	6.75 (4.27)	3.45 (0.19)	48.14 (1.02)	2.25 (0.50)	22.57 (6.68)	0.13 (0.07)	
Teufels-seemoor	10	36.10 (8.58)	3.36 (0.17)*	51.69 (1.93)*	1.96 (0.14)*	26.46 (2.48)*	n.s.	
Note:

Mean values, standard deviations (SD, in brackets) and respective number of replicas for thickness and the soil parameters pH value, TOC, TN, C/N ratio and BD of the newly formed peat moss peat above the degraded peat of the study sites Müllerwiese, Seggenkuten and Teufelsseemoor in 2021. Asterisks indicate significant differences (p < 0.05) between newly formed peat and degraded peat beneath after restoration.

Discussion

There are only a few comprehensive studies in North-East Germany regarding values for TOC in degraded peat soils. Succow (1988) determines a mean of 35.0% (span: 7.2 to 49.6, standard deviation: 12.5) for earthified peats. Kühn et al. (2015) find a comparable mean of 31.4% for earthified peats and a lower mean of 27.5% for murshified peats, since organic carbon decreases with increasing soil aeration and degradation. With the exception of Beerenwiese, the mean values for TOC in the study sites before restoration are higher than the comparative values given in literature. This is probably due to the fact, that these fens have been drained only moderately during usage as extensive grassland and therefore, there was only slight topsoil degradation and loss of organic carbon in most of the study sites. Three of the six fens showed only a beginning earthification before rewetting (Table 1). The non-organic mass of the six fens is mainly attributable to the loss of organic content during peat degradation since there are no surface waters carrying in allochthonous mineral material. Furthermore, the above ground catchment areas have been forested for the past decades, so that soil erosion into the fens is negligible.

The most comprehensive compilation of Holocene peat soil properties for the northern hemisphere is given by Loisel et al. (2014), whereby ombrotrophic bogs are more strongly represented than minerotrophic fens (20% of all sites), only few plots are located in Germany, and the focus is on complete peat cores from surface to peatland base. Loisel et al. (2014) determine a mean TOC of 47.4% (n = 96) for ‘humified’ peat, as highly decomposed peat which is not assignable to a specific plant group. Succow (1988) found a mean of 40.8% TOC for non-degraded sedge peat in North-East Germany. The TOC values for sedge-dominated sites (Beerenwiese, Boberowseewiese, Jägerwiese) in the near-surface peats are higher (42.5% to 50.4%) about 17 years after restoration. Loisel et al. (2014) give a mean of 50.5% TOC in northern peatlands (n = 519) regarding ‘herbaceous’ peat (including peats formed by sedge roots), which is reached by one of the sites in 2019–2021. For peat moss peat, Succow (1988) found a mean of 48.6% TOC. Values for degraded near-surface peats in peat moss-dominated sites are comparable for two sites (Müllerwiese: 47.7%, Seggenkuten: 47.6%) and are higher in Teufelsseemoor: 54.0% in 2019–2021. All three sites show higher TOC than the mean value for ‘Sphagnum’ peats in the northern hemisphere: 46.0 (n = 1520) as given by Loisel et al. (2014) after restoration. But, the comparison of TOC of the near surface peat with TOC of deeper layers must be interpreted cautiously, as the deeper layers have been undergoing continuous decomposition and the newly added material and TOC will only partially or even never become part of the long-term carbon store (cf. Young et al., 2019).

The distinct increase of TOC in five study sites (four significantly, whereof three sedge-dominated, one peat moss-dominated) after rewetting as well as the comparison with carbon contents of non-degraded sedge and peat moss peat indicate that only a few decades after restoration, an organic carbon content typical for undisturbed peatlands can be regained in degraded peat horizons. Prerequisite for these developments is a water table typical for near-natural mires allowing for accumulation of peat and the re-establishment of potentially peat forming plant species. These potentially peat forming species, such as Sphagum spec. and most Carex species are not only dependent on wet conditions to be competitive, but a high water table is a requirement to allow dying plant parts of these species to form peat and not be oxidated. The divergent development of TOC in Teufelsseemoor and Müllerwiese emphasizes the impact of the hydrological conditions on carbon storage: Teufelsseemoor, with the highest increase in the water table, shows the highest gain in TOC in near-surface peats after restoration, whereas Müllerwiese, with a water table mainly still below the peatland surface and ongoing aeration of peat, shows no gain in TOC.

Research on the restoration of fen systems indicates, that the lower the degree of initial degradation, the higher the potential for restoring fens to near natural conditions (cf. Grootjans & Van Diggelen, 1995; Klimkowska et al., 2010). However, research so far focused mainly on the development of typical, sometimes highly endangered plant species, not on soil properties. Our results do not indicate that a lower initial degree of soil degradation leads to a higher restoration success concerning TOC, since we determined a significant gain in TOC in sites with a beginning earthification (Jägerwiese, Teufelsseemoor), an earthification (Beerenwiese) as well as a murshification (Boberowseewiese) before restoration. We recommend additional research with a higher number of study sites with different soil degradation stages to further investigate this debate.

Regarding sedge-dominated fens, the input of new carbon happens by ingrowing and later dying sedge roots (Michaelis, Mrotzek & Couwenberg, 2020). In the peat moss-dominated fens of our study, new peat of dying peat mosses is deposited on top of the degraded peat. A possible explanation for carbon enrichment in lower soil horizons may be the deposition of dissolved organic carbon released in case of high water tables. Mrotzek et al. (2020) and Michaelis, Mrotzek & Couwenberg (2020) report a newly deposited material after 20 years of restoration of roots, radicals and litter, and describe it as a first accumulation of litter leading to a so-called ‘proto-peat’. This indicates that also in sedge-dominated peatlands, new deposits on degraded peat soil are possible within a time-frame comparable to our study.

Still, TOC values of our study within the near-surface peats cannot be directly transferred to determine long-term carbon storage. Due to ongoing decay in the highly dynamic, partially not water saturated near-surface peat (‘acrotelm’), our results can only be used to describe the potential carbon to be transferred into the permanently water saturated, long term carbon store (‘catotelm’), where decomposition is very slow (Tolonen & Turunen, 1996; Young et al., 2019). This process is influenced by various factors such as climate, water table, vegetation and peatland type (cf. Tolonen & Turunen, 1996; Malmer, Svensson & Wallén, 1994; Charman et al., 2015; Milner et al., 2020). Young et al. (2019) show, that using changes in carbon accumulation in near-surface peats for estimating long-term carbon storage might be misleading and should not be used as the only source of evidence of management strategies on peatland carbon. Especially in the study area, due to the climatic change with higher temperatures and decreased precipitation, negative climatic water balance (cf. DWD (Deutscher Wetterdienst), 2022) and therefore high dependency of the peatlands on inflowing water from the surrounding environment, the development of the carbon transferred into long-term-store is unclear. However, the water tables in our study sites are relatively stable after restoration also in years of low precipitation as in 2018. This could be a hint for increasing resilience of the peatland systems after restoration by increasing porosity and water storage capacity of the near-surface peats.

Regarding content of TN, there are also only few comprehensive studies in Germany. Grosse-Brauckmann (1990) and Naucke (1990) state values of 2.0% to 4.0% and respectively 2.5% to 4.5% for upper fen soils in general. Feige (1977) gives contents of 2.47% to 3.07% for undisturbed sedge peat and 0.5% to 1.33% for undisturbed peat moss peat. Loisel et al. (2014) give for northern peatlands a mean of 1.5% (n = 96) for ‘humified’ peat, a mean of 1.7% (n = 518) for ‘herbaceous’ peat and a mean of 0.7 (n = 1,523) for ‘Sphagnum’ peat. TN values of our study sites are in line with comparative values in Germany but higher than values given for the northern hemisphere, presumably due to the high share of ombrotrophic bogs having lower TN contents than fens which are supplied with mineral water. The detected relative TN decrease in four sites (only one significant) after rewetting can be mainly attributed to the gain of carbon, as described by Malmer & Holm (1984), Ohlson & Økland (1998) or Turunen, Tahvanainen & Tolonen (2001). Malmer & Wallén (2004) assume that nitrogen is conserved quantitatively in the peat because of hardly any losses during decay. This is also apparent regarding the C/N ratio. The ratio increases in four sites (three significantly), indicating a lowered nutrient availability. Again, this can be mainly attributed to the gain of carbon after rewetting. These findings are of high importance for specific fen plant species, with a high share of adaption to rather mesotrophic conditions.

Mean values for BD in North-East Germany are given by Kühn et al. (2015) for earthified peats: 0.3 g/cm3 and murshified peats: 0.5 g/cm3 as well as by Schindler, Behrendt & Müller (2003) for earthified peats: 0.28 g/cm3 and murshified peats: 0.42 g/cm3. Values are comparable to our sites before restoration, although some sites show a lower BD due to an only beginning earthification. The significant decrease of BD after rewetting in four sites can be attributed to the gain in carbon, as there is a clear relation between low BD and high carbon content (Chapman et al., 2017; Wittnebel, Tiemeyer & Dettmann, 2021). The most remarkable decrease of BD in Boberowseewiese, in particular, can only be fully explained by the combination of the carbon increase due to ingrowing and dying sedge roots and the loosening of peat by buoyancy after resaturation. Values after restoration in the sedge-dominated sites are in line with values given by Loisel et al. (2014) for ‘herbaceous’ peat in northern peatlands with a mean of 0.12 g/cm3 (n = 3,188). The values for our peat moss-dominated sites are about double as the mean of 0.08 g/cm3 (n = 4,372) for ‘Sphagnum’ peat given by Loisel et al. (2014).

pH values remain relatively stable before and after restoration, since quantity but not quality of the feeding water has changed by rewetting.

Conclusions

Fen restoration with re-establishment of water tables typical for undisturbed peatlands, with regard to our six study sites in North-East Germany, leads to stable pH values, increased TOC comparable to contents of undisturbed peatlands, decreased nutrient availability in terms of C/N ratio, and decreased BD (resembling values of undisturbed peat for sedge-dominated sites) in the degraded, near-surface peats.

Further, as for mire-specific biodiversity described by Hammerich et al. (2022) not only on the level of species, but with peat accumulation and water table close to surface as determining eco-hydrological properties on the ecosystem level, the regeneration of the near-surface peat can also be seen as central for re-establishing the biodiversity of mires. We assume that along with the rise in water table, the decrease in nutrient availability enhances the potential for the recolonization by mire-specific plant species, which are not only adapted to water-saturation but often extreme pH values and nutrient conditions (cf. Minayeva et al., 2008). The decrease of BD and the establishment of a new undisturbed peat layer and thereby a developing functioning acrotelm are described as a key aim in the restoration of peatlands (Convention on Wetlands, 2021) to lead to a self-regulating system.

Concluding, the gains in carbon content are a central argument promoting peatland restoration in terms of climate change mitigation. Further, peatland restoration can favor the establishment of mire-specific biodiversity and support the redevelopment of ecosystem functions similar to undisturbed peatlands.

Supplemental Information

Supplemental Information 1 Readme for raw data.

Supplemental Information 2 R-Script.

Supplemental Information 3 The data on the newly deposited peat layers above the old degraded peat.

Supplemental Information 4 The data on water table measurements for all study sites.

Supplemental Information 5 The data of all soil parameters with values of before restoration (2002/04) and after (2019/21).

Supplemental Information 6 The data on differences in values from before to after restoration of all soil parameters.

The authors thank the administration of the nature park “Stechlin Ruppiner Land” for good collaboration over the years. We thank all who provided thoughtful reviews and comments.

Additional Information and Declarations

Competing Interests

Author Contributions

Field Study Permissions

Data Availability

Jenny Hammerich, Corinna Schulz, Robert Probst and Vera Luthardt declare that they have no competing interests. Thomas Lüdicke is employed by UWEG Engineers & Analytics GmbH.

Jenny Hammerich conceived and designed the experiments, performed the experiments, analyzed the data, prepared figures and/or tables, authored or reviewed drafts of the article and approved the final draft.

Corinna Schulz conceived and designed the experiments, performed the experiments, analyzed the data, prepared figures and/or tables, authored or reviewed drafts of the article and approved the final draft.

Robert Probst conceived and designed the experiments, performed the experiments, analyzed the data, prepared figures and/or tables, and approved the final draft.

Thomas Lüdicke performed the experiments, prepared figures and/or tables, and approved the final draft.

Vera Luthardt conceived and designed the experiments, authored or reviewed drafts of the article, and approved the final draft.

The following information was supplied relating to field study approvals (i.e., approving body and any reference numbers):

Field experiments were approved by the administration of nature park ‘Stechlin-Ruppiner Land’, belonging to the State Environmental Agency of Brandenburg.

The following information was supplied regarding data availability:

The script and raw data are available in the Supplemental Files.

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
