# Peer review of "Carbon content and other soil properties of near-surface peats before and after peatland restoration"

_PeerJ, doi:10.7717/peerj.17113_

## Round 0.1 · original submission · Major Revisions

The scientific article is devoted to the study of issues of peatland restoration relevant for modern science, among the variety of functions of which carbon storage occupies an important place. The obtained results of the study are characterized by significant validity and relevance in the conditions of climate change, an increase in the carbon content in the atmosphere on a global scale. The authors conducted a qualitative systematic field survey of peatlands in six restored bogs in Northeast Germany and concluded that peatland restoration contributes to the restoration of peatlands as carbon sinks.

It is also necessary to note the qualitative structuring of the article and the practical value of its main results. The manuscript follows the direction of the journal PeerJ. In order to improve the quality of the manuscript, I recommend, if possible, taking into account the comments of the reviewers, which you consider appropriate.

**Language Note:** PeerJ staff have identified that the English language needs to be improved. When you prepare your next revision, please either (i) have a colleague who is proficient in English and familiar with the subject matter review your manuscript, or (ii) contact a professional editing service to review your manuscript. PeerJ can provide language editing services - you can contact us at copyediting@peerj.com for pricing (be sure to provide your manuscript number and title). – PeerJ Staff

·

Basic reporting

The article is professionally structured and written.

1. Structure and field background / context

The Introduction could provide more background regarding the significance of total organic carbon content (TOC) and C/N ratio, which turn out to be important when evaluating the success of restoration in the Discussion. Then, when readers reach the results on TOC and C/N ratios starting on line 188 they will have the context to appreciate the significance of the results.

2. Line 26: "... six restored fens in North-East Germany before (2002/04) and after (2019/21) restoration."

The stroke ("/") for date range creates ambiguity because the stroke is also used in English to mark alternatives, and as a date field separator, so when I first read 2002/04 I am not completely sure whether I should understand "April of 2002", "either 2002 or 2004", or "2003 through 2004". To eliminate this ambiguity, I would suggest replacing this throughout the paper with a more explicit notation. One possibility could be to use an en dash to indicate a range and write out the dates in full: I think "before (2002–2004) and after (2019–2021)..." would be unambiguous. Or, the ranges could be written in words: "before (2002 through 2004) and after (2019 through 2021)..."

This is also relevant to data file Statistics_R_Stechlin_final.csv.

3. Lines 34-35: "The structure of the formerly degraded earthified or murshified peat horizons changed after restoration..."

I was initially mystified by the terms "earthified" and "murshified," so I looked them up in the 13-page index of _Mires and peatlands of Europe_. Not finding them, I consulted a German colleague and searched the literature. It seems to me that these words are not in widespread use in the English-language scientific literature apart from some papers by the authors, notably Schulz et al. (2019; 10.19189/MaP.2019.OMB.StA.1817). From the description there, I have the impression that, like von Post decomposition classes, these are semi-qualitative descriptors that require some direct experience to really understand them, but, also like von Post classes, it should be possible to provide some qualitative descriptions that gives readers a sense of what they mean.

In my view, one of the most effective ways to make this article more accessible to a wider audience would be (1) ideally, to replace these words with descriptive phrases; and (2) to carefully define these words or phrases on their first use.

On lines 204-205 I find these potentially useful phrases: "(crumb grain structure for earthified peat and fine granular structure for murshified peat according to Schulz et al. 2019)". Perhaps "crumbified peat" and "granulated peat", with full definitions?

If the use of "earthified" and "murshified" seems to be unavoidable in the main text, I would recommend defining them carefully on their first use and avoiding them in the Abstract to make the paper more approachable to a wider readership.

4. Lines 209-210: "The mean thickness of this layer ranges from 6.8 cm in Seggenkuten to 36.1 cm in Teufelsseemoor."

Consider providing accumulation rates (cm / y)?

6. Lines 222-282, Discussion

The first paragraph of the Discussion appears to be very long, but I am not sure, because there is neither additional space nor indentation to mark the beginnings of paragraphs.

At Line 244, the sentence beginning with "The distinct increase of TOC in five study sites..." is important; consider beginning a new paragraph here to draw attention to this result?

Similarly, I was not sure if the sentence starting on line 259 ("Still, TOC values of our study within the near-surface peats cannot be directly transferred to long-term carbon storage...") is intended to start a new paragraph, but this would be another natural place to split this section of text.

7. Lines 224-226: "Kühn et al. (2015) find a comparable mean of 31.4 % for earthified peats and a lower mean of 27.5 % for murshified peats, since organic carbon decreases with increasing soil aeration and degradation."

Consider discussing briefly the composition of the non-OC mass. Is this partly allocthonous mineral content carried in by surface waters, or perhaps contributed by some form of soil amendment associated with agriculture, or is the higher mineral content entirely attributable to the loss of organic content during peat wastage?

8. Lines 226-227: "With the exception of Beerenwiese, the mean values for TOC in the six study sites before restoration are higher than the comparative values,... "

This is not apparent in Figure 5... Although Figure 5 (presumably) shows quantiles and the mean is not shown, TOC appears to be higher after restoration at all sites except for possibly Seggenkuten?

9. Lines 278-279: "The detected relative TN decrease in four sites (only one significant) after rewetting can be mainly attributed to the gain of carbon,..."

This should be apparent in the C/N ratio, discussed next; consider discussing these together?

10. Figure 2

These are great pictures. If it is easy, the site name labels could be made more legible by placing a black rectangle behind the white lettering, or a white rectangle behind black lettering.

11. Figure 3

More great pictures.

*Minor issues of ambiguous or imprecise wording*

12. Lines 39-40: "TOC of the near-surface peats increased comparably to values in undisturbed peatlands"

"increased comparably" is ambiguous

13. Line 50: "... a naturally accumulated layer of peat which is sedentarily accumulated material"

A comma is required: "peat, which is sedentarily..."; or "peat, or sedentarily..."

14. Line 53 and 263: "... the decomposition of the on-site dead biomass is very low..." and "... where decomposition is very low..."

Consider "is very slow"

15. Lines 62-63: "This results in an estimated total amount of 6.3 million t CO2-Eq. per year"

Consider "estimated total emissions" in place of "estimated total amount"

16. Lines 64-65: "The once existing specific biodiversity of mires"

Consider "biota" in place of "biodiversity?" "Biodiversity" is often used to mean a (possibly quantitative) characterization of the amount of variability among organisms in a region.

17. Lines 71-73: "In order to reduce CO2 emissions and restore the function as carbon sinks, as well as biodiversity and other ecosystem functions of mires, peatland restoration efforts have risen"

"efforts have risen": Is it possible to be more specific about the time frame over which the shift from exploitation towards restoration of peatlands has occurred in Europe?

18. Lines 74-77: "Estimating reinitiated carbon storage in near-surface peats of restored peatlands... is a cornerstone in the recognition and role of peatlands as carbon sinks"

What is being estimated here? Are we determining whether or not, or perhaps when, restored peatlands have become net sinks again? Are we estimating the sink strength (amount of carbon sequestered per time)?

The combination of "recognition and role" is part of what confuses me here. I can see how determining whether a peatland is sequestering carbon is critical to recognizing the peatland as a carbon sink. But this determination is not really relevant to the role of the peatland as a carbon sink per se---the peatland is, or isn't, a carbon sink, whether we can determine that it is taking up carbon or not.

Splitting this into two sentences might help?

19. Line 93: "high relief energy": define

20. Line 94: "diversity hydrogenetic peatland types"

This phrase will not be familiar to non-specialists. Is there a way to reorganize this paragraph so that hydrogenetic peatland types are defined (and motivated) before the use of this phrase?

21. Line 104-105: "Regarding the trophic conditions, which are graduated by the C/N ratio of the peat..."

Consider replacing "graduated by" with "quantified by" or "scored by"

22. Lines 111-112: "The climatic water balance is negative, so that the water content of the peatlands is always dependent on inflowing water from the surrounding environment."

Reference?

23. Line 113: "Until 2002, all fens were drained"

Consider "all six fens" to make it clear that this statement refers to the fens in this study (and not all fens globally)

24. Line 115: "Water tables in all fens were several dm below peatland surface."

Consider providing the mean water table position for the fens, in cm, in preference to dm.

25. Figure 1

The caption states that the units on the vertical axis are dm, while the vertical axis is labeled "[cm related to surface]". I think the correct units are dm (because otherwise the fluctuations would be extremely small), but I suggest using cm, which are in more widespread use and are used elsewhere in the document (in the units for bulk density, for example).

26. Lines 137-138: "Sampling plots were recorded via precision-GPS (LEICA GS 50) to be relocated for future sampling"

"relocated" is often used to mean "moved." An easy solution here is to simply remove the clause (something like: "Sampling plot locations were recorded via precision-GPS (LEICA GS 50)").

27. Line 195: "The C/N ratio as key parameter for trophy" Replace "trophy" with "trophic conditions" (as used on line 102) or "tropic status"

28. Line 211: "analogue": "analogous to"

29. Line 212: "filthy"

Define, or replace? I wasn't sure what was meant here, the word in colloquial English is evocative but informal and has a negative connotation.

30. Line 228: "due to the only extensive land use and drainage"

Expand; the meaning is unclear.

31. Lines 291-293: "The significant decrease of BD after rewetting in four sites can be attributed to new water saturation and thereby buoyancy and loosening of peat"

Wording is confusing; perhaps "... can be attributed to the loosening of peat by buoyancy after resaturation"

*Raw data*

32. Raw data: Where are the units defined for the data? Consider a separate README file to describe the data?

33. Raw data: The comma (,) is used as a decimal separator / radix point in the data files, while a point (.) is used in the text. I have no objection to the comma but it conflicts with the use of the decimal point in the text, and with the convention in most English-speaking countries outside South Africa. Consider switching to point, or including a comment about the use of the comma in the README?

34. Raw data: "Profil": replace with "profile"?

35. In peat_old_new.csv: Is the file content meant to start with "ü"?

36. In Statistics_R_Stechlin_final.csv: What do the numbers in the column "new_peat" mean? Include in README?

37. In Statistics_Waterlevel.csv: Consider using ISO 8601-compliant dates (2006-12...)

38. In Statistics_waterlevel.csv: waterlevel - what do missing values mean? Could these months perhaps be simply excluded from the table?

Experimental design

The research question is well designed and fills an identified knowledge gap. Methods are mostly described in sufficient detail to be replicated.

39. Figure 1 and Methods

How was the water table position measured, and how frequently? If there was any microtopographic relief in the soil surface, what was the reference elevation for water table measurements (hummocks? hollows?)?

40. Lines 150-152: "For Teufelsseemoor, volumetrically sampling was not possible due to the distinct fluffiness of the newly formed peat."

Why not? Was it difficult to cut the peat without distorting its shape?

Validity of the findings

Underlying data have been provided. Conclusions are for the most part well stated, linked to the original research question, and supported by the results.

41. However, there is one exception, in the Conclusions, at lines 309-313: "the regeneration of the near-surface peat can also be seen as central for re-establishing the biodiversity of mires. We argue that along with the rise in water table, the decrease in nutrient availability enhances the potential for the recolonisation by mire-specific plant species, which are not only adapted to water-saturation but often extreme pH values and nutrient conditions"

I would agree with these statements but they are not directly supported by the results here, and therefore they are not conclusions of this study.

Reviewer 2 ·

Basic reporting

No comment

Experimental design

1) It does not appear that a core was sampled down through each of the peats from the different sites before and after. I would have thought that was critical when the conclusions are based on making comparisons before and after restoration. It is stated that there was a “distinction regarding sampling depth” for the different types of peatland site. This seems to me introducing an element of choosing what to look at which is problematic for subsequent comparisons.

Validity of the findings

2) At no point in the paper is there any mention of the possible presence of mineral matter, and yet it can be an indicator for distinguishing various types of murshes. Also, higher bulk density (BD) values are associated with higher ash content, see S.J. Chapman et al Mires and Peat, Volume 19 (2017), Article 23, 1–11 “Refining pedotransfer functions for estimating peat bulk density”. This is relevant as this paper quotes high bulk density values for some of the soils and in one case notes (line 201) a change from 0.48 to 0.15 g/cm3, after rewetting and attributes reductions in BD to “new water saturation and thereby buoyancy and loosening of peat” (line 292-293). I am not convinced that this is in fact a change and not measurement of new type of peat formed.

3) As alluded to in the comment above, there does often seem to be confusion as to whether comparisons described are changes to an existing peat or changes from an existing peat to the newly formed peat. Again, for reliable comparisons having this established clearly is critical. There are several points in the paper where this could be clarified by the authors.

Additional comments

4) Although the authors do reference Young et al (2019) I feel they haven’t fully discussed their results in relation to the findings of that paper. And this would be a helpful addition.

---

## Round 0.2 · Minor Revisions

The submitted manuscript is significantly improved and refined, compared to the previous version. The authors responded to the main remarks and suggestions of the reviewers, which improved the quality of the manuscript. Regarding this manuscript, minor remarks remained.

·

Basic reporting

Minor comments:
- Line 85: "we demonstrate the change regarding the soil parameters...": Suggest changing "we demonstrate the change regarding" to "we report changes in"
- Line 92: "..., which is seen as the widely accepted relation between these soil parameters on a global scale": I wasn't sure what was meant here; perhaps this phrase could be removed?
- Lines 135-136: "near-surface peats in all sites exhibited different stages of oxygen-induced soil degradation of primarily accumulated peats." I think I know what is meant here, but the phrase "primarily accumulated peats" is ambiguous in English because "primarily" typically is used to mean "mostly." I suggest simplifying this to "near-surface peats in all sites exhibited different stages of oxygen-induced degradation," or otherwise expanding this into two sentences to make the meaning clear.
- Lines 136-142: Nice introduction to "earthified" and "murshified," this resolves the issue mentioned in my previous review.
- Lines 239-240: "Mean amounts of TOC are comparable to amounts of soil beneath after restoration for two of the three sites, ..." Meaning is a little unclear. Perhaps "Mean TOC in new peat is comparable to TOC in older peat at two of the three sites after restoration, ..." (if this is what was intended?).
- Lines 275-279: "the comparison of TOC of the near surface peat with TOC of deeper layers must be interpreted cautiously, as the deeper layers have been undergoing slow, still constant decomposition and the newly added material and TOC will not entirely, if at all, become part of the long-term carbon store (cf. Young et al., 2019)." This is an important point. I suggest minor revision for clarity: consider replacing "undergoing slow, still constant decomposition" with "undergoing continuous decomposition" and perhaps replacing "newly added material and TOC will not entirely, if at all, become part of the long-term carbon store" with something like "much or all of the newly added material and TOC may never become part of the long-term carbon store" or "only some, or perhaps none, of the newly added material and TOC will become part of the long-term carbon store."
- Lines 331-333: "The most remarkable decrease of BD in Boberowseewiese, in particular, can not only be explained by the carbon increase due to ingrowing and dying sedge roots, but additionally by the loosening of peat by buoyancy after resaturation." I am a little unsure of what is meant here, because the phrasing is somewhat unusual; perhaps consider the following, if it matches the intended meaning: "..., in particular, can only be fully explained by a combination of the carbon increase due to ingrowing and dying sedge roots and the loosening of peat by buoyancy after resaturation"
- The added references and additional explanation in the Introduction and Discussion are interesting and greatly enrich the paper.

- Minor comments on the raw data and description:
- In the README, there is a minor typo on the first page: "... in all table oft he raw data."
- For a global audience, I suggest stating in the README that the CSV files use ISO-8859-1 (Latin 1) encoding for text.

Experimental design

No additional comments.

Validity of the findings

No additional comments.

Additional comments

This revision fully addresses my previous comments. I find the manuscript much improved by the additional references and context provided by the authors.

Reviewer 3 ·

Basic reporting

Comment 1: Introduction (generally)
I agree with reviewer 1 to add more background information concerning TOC and C/N. However, I guess that doing this at the end of the introduction is unusual and not a good idea. I propose, to shift the last two paragraphs (lines 90-104) and set them behind line 78. Furthermore, emphassise and highlight the novelty and innovation of the study in the last paragraph by referring to different hydrological and ecological site conditions and degradation stages.
Comment 2:
Line 51 (introduction): add „according to the German definition of mires (Adhoc AG Boden 2005)“ behind „30% dead organic material“.
Comment 3:
Line 69 (introduction): water storage basis or „basin“ or water storage capacity?
Comment 4:
Lines 113-118 (study sites): To my mind it is not necessary to describe the hydrogenetic types defined by Succow (1988) in general. Thus, delete these sentences and change the line 119 in „The specifications regarding the „hydrogenetic and ecological peatland type“ given in table 1 follow Succow (1988). Furthermore, describe shortly in the following the investigated six sites by referring also to the own measured data of pH and C/N-ratio in figures 4 and 7. As a result, the sites can easily divided into two subneutral and four acidic types and further into two eutrophic (Beerenwiese, Boberow Seewiese), two mesotrophic (Jägerwiese, Teufelsseemoor) and two meso- to eutrophic sites (Müllerwiese, Seggenkuhlen). Please mention additionally the variety of hydrogenetic types (terrestrialisation and percolation fens, kettle hole peatlands etc.)
Comment 5:
I agree with the next two paragraphs (Laboratory and statistical analysis)
Comment 6:
I urgently miss at the beginning of the chaper results a paragraph dealing with the hydrological conditions after restoration by referring to figure 1. I think, there are distinct differences between the investigated peatlands which potentially influences restoration success, e.g. habitat quality for plant species.
Comment 7:
I propose also to start the discussion with interpreting the hydrological development of the peatlands. For me it is e.g. unclear why rising water tables have been observed during the process of restoration. What are the reasons (e.g. upwelling of peat)? By having a look on the photo of „Teufelsseemoor“ in figure 2, there is no indication for a shallow lake as can be expected by the soil water dynamic (up to 75cm above soil surface) in figure 1. Furthermore, are there any reasons for explaining the low restoration success of the peatland „Müllerwiese“ due to relative low water tables.
Comment 8:
Lines 254 ff. (discussion). The authors explain the relative high TOC-values before restoration with the comparatively „only slight topsoil degradation“. Nevertheless, they measured at least partly significant increases TOC despite of this initially high values. However, what does that mean for the potential success of restoration managament in fen ecosystems and landscape planning? Is a low degradation stage (in comparison to highly degraded initial situations) a indispensible prerequisite. Furthermore, at this point there is a big chance to point out the necessity for analysing the degradation stage of peatlands more in detail as have been realised in this study. This could be also mentioned in the chapter conclusions.
Comment 9:
Lines 284 ff. (discussion): Please discuss at this point more in detail the preconditions of specific (e.g. peat-forming) species by using also the hydrological results.
Comment 10
Lines 305 ff. (discussion): The statement concerning unclear development of peatland functioning due to climate change is comprehensible, but however, by comparing the water table dynamics of the restored peatlands in this study I see no distinct decreases of water tables also in years with very low precipittaion as e.g. in 2018. I guess, this could be a hint for increasing resilience of the systems after restoration.

Experimental design

no comment

Validity of the findings

no comment

Additional comments

Peatland restoration is currently worldwide one of the most important ecological solutions to deal with the climate crisis. Although a start of restoration was made several decades ago the first activities in fen ecosystems of Central Europe were mainly aimed at preserving and developing biodiversity and retaining nutrients whereas the function of a carbon store only became priority in recent years. As a consequence, there is a large gap in scientifically based results on long-term changes in soil chemical and physical parameters of fens as a result of rewetting measures. Therefore, the results of this study are of high importance in order to reduce this shortcoming. The advantages of the investigations are firstly the availability of data before restoration and secondly the variety of sites concerning ecological and hydrological condition as well as differences in their degradatation stage.

---

## Round 0.3 · accepted · Accept

The updated version of the submitted manuscript has been substantially revised and is at a much higher level. The presented results are substantiated, the presentation of the material is concise. The reviewers confirmed the completeness of the scientific material and the readiness of the manuscript for acceptance, which was confirmed by their decisions.

Reviewer 3 ·

Basic reporting

Ready for publication

Experimental design

no comment

Validity of the findings

no comment

Additional comments

General (final) evaluation:
After reading the corrections of the authors in the latest version of the manuscript, I fully agree with the remark of the first reviewer, that the paper is now ready for publication (excluded final editorial corrections which I’m not able to recognize). To my mind, all recommendations of the reviewers have been considered and I agree again with my colleague, that the required new supplements and explanations significantly enhance the quality of the paper. Finally, I would like to emphasize once again its unique character and high importance for an international audience.
Congragulations and best regards